# An Assessment of Long-Term Physical and Emotional Quality of Life of Persons Injured on 9/11/2001

**DOI:** 10.3390/ijerph16061054

**Published:** 2019-03-23

**Authors:** Robert M. Brackbill, Howard E. Alper, Patricia Frazier, Lisa M. Gargano, Melanie H. Jacobson, Adrienne Solomon

**Affiliations:** 1World Trade Center Registry, New York Department of Health and Mental Hygiene, New York, NY 10013, USA; halper@health.nyc.gov (H.E.A.); lgargano1@health.nyc.gov (L.M.G.); mjacobson@health.nyc.gov (M.H.J.); asolomon1@health.nyc.gov (A.S.); 2Department of Psychology, University of Minnesota, Minneapolis, MN 55455, USA; Pfraz@umn.edu

**Keywords:** injury, physical health, mental health, World Trade Center disaster, Short Form-12 (SF-12), HQoL, 9/11

## Abstract

Fifteen years after the disaster, the World Trade Center Health Registry (Registry) conducted The Health and Quality of Life Survey (HQoL) assessing physical and mental health status among those who reported sustaining an injury on 11 September 2001 compared with non-injured persons. Summary scores derived from the Short Form-12 served as study outcomes. United States (US) population estimates on the Physical Component Score (PCS-12) and Mental Component Score (MCS-12) were compared with scores from the HQoL and were stratified by Post-traumatic Stress Disorder (PTSD) and injury status. Linear regression models were used to estimate the association between both injury severity and PTSD and PCS-12 and MCS-12 scores. Level of injury severity and PTSD history significantly predicted poorer physical health (mean PCS-12). There was no significant difference between injury severity level and mental health (mean MCS-12). Controlling for other factors, having PTSD symptoms after 9/11 predicted a nearly 10-point difference in mean MCS-12 compared with never having PTSD. Injury severity and PTSD showed additive effects on physical and mental health status. Injury on 9/11 and a PTSD history were each associated with long-term decrements in physical health status. Injury did not predict long-term decrements in one’s mental health status. Although it is unknown whether physical wounds of the injury healed, our results suggest that traumatic injuries appear to have a lasting negative effect on perceived physical functioning.

## 1. Introduction

The World Trade Center (WTC) disaster on 11 September 2001 exposed thousands of persons to both environmental pollutants and psychological trauma, which have had long-term physical and psychological ramifications. Having been injured on 9/11 has emerged as a common risk factor for both physical and mental health conditions [1,2,3]. For example, one study found that, among those injured on the day of the 9/11 disaster, the likelihood of having been diagnosed with a physical health condition, including respiratory disease problems (i.e., asthma, chronic bronchitis, emphysema) and/or circulatory disease (i.e., heart attack, angina, stroke) increased with the number of types of injuries they sustained (e.g., burn, head injury, and musculoskeletal) [4]. Injury on 9/11 also increased the likelihood of posttraumatic stress disorder (PTSD) 2-fold to 3-fold, which was measured by a post-traumatic stress checklist (PCL) screening instrument after adjusting for demographic factors and other WTC-related exposures [2,3]. In addition, physical injury has been reported as a risk factor for psychological sequelae among persons directly affected by other disasters, such as Hurricane Ike [5] and the 2004 Southeast Asia tsunami [6]. 

In general, among those who experience serious injuries, psychological distress related to anxiety, depression, or post-traumatic stress disorder (PTSD) can persist for years after the injury [6]. For instance, in a meta-analysis of the long-term impacts of injuries incurred in motor vehicle crashes, ranging from serious spinal cord injuries to less life-threatening musculoskeletal injuries, showed that the injuries had large impacts on psychological distress that, in some cases, increased in magnitude over time [7]. 

Serious injuries can also result in increased vulnerability to physical health conditions in the absence of psychological pathology [8]. This suggests that an injury, in addition to its immediate physical damage, can have long-term effects on physical health without an intervening psychological factor. 

Given that mortality from injury has declined substantially because of treatment advances, the primary focus for non-fatal injuries currently is the impact of injuries on functionality and quality of life. For instance, Danish patients injured between 1995 and 2005 were more likely than a non-injured group of people to experience poor/very poor health [9]. In the Netherlands, from 1999 to 2000, limitations in mobility, self-care, and the ability to conduct daily activities were self-reported by severely injured adult patients followed up between 12 and 18 months post-injury [10]. Long-term declines in overall physical and mental health status following trauma were found in studies in Oslo, Australia, and the United States (US) using standard instruments (e.g., Short Form 36, Brief Pain Inventory, and Short Form 12) [11,12,13]). In the Danish study, Toft observed the effects on overall health up to 10 years after the injury [9] and for at least 2 years for both physical and mental health when compared with national norms [12,14]. In most cases, a greater relative decline in physical health compared with mental health was observed. However, researchers typically do not include in their assessment the relationship between PTSD and functional health status, especially physical health, nor do they employ a non-injury comparison group.

A prior qualitative inquiry of persons who sustained an injury on 9/11 found injured persons reported that their injuries were debilitating and limited daily activities, especially musculoskeletal injuries that required multiple surgeries with physical therapy [15]. In addition, quality of life and social integration problems emerged among participants in the study regardless of whether they had a history of PTSD [15]. 

This study examines the long-term effects of injury sustained on 9/11, including functional effects and a comparison group without injury. Based on the findings from the qualitative study, we hypothesized that the severity of the 9/11 injury would predict deficits in health status for both physical and mental health domains. We also hypothesized that PTSD history would be sufficient —but not a necessary factor—in observed long-term detriments in physical and mental health, as measured by the Short Form-12 (SF-12). In addition, it was hypothesized that the presence of other factors, such as social support and self-efficacy, would ameliorate the impact of injury and PTSD on physical and mental health status.

## 2. Materials and Methods

The World Trade Center Health Registry (Registry) is a prospective cohort that monitors the physical and mental health of 71,426 persons exposed to the attacks on 11 September 2001. The populations at risk included rescue/recovery workers who were involved in rescue, recovery, and disaster clean-up from 11 September 2001 to 30 June 2002. This includes residents who lived south of Canal Street in lower Manhattan on 9/11, persons who worked and were present south of Canal Street on 9/11, passersby and others who were occupants of destroyed or damaged buildings on 9/11, and students who were registered at, or staff employed by, schools located south of Canal Street. The Registry enrolled 17% of an estimated 409,492 persons in the populations-at-risk identified by the Registry [16], which varied from 34% of rescue/recovery workers to 11% of passersby. Since the enrollment survey in 2003–2004 (Wave 1), the Registry has conducted three health surveys: Wave 2 in 2006–2007, Wave 3 in 2011–2012, and Wave 4 in 2015–2016 (See references 2 and 17 for more detail on methods and participation) [2,17,18]. The Registry was approved by the New York City Institutional Review Board and Federal Centers for Disease Control and Prevention (15005).

### 2.1. Study Sample 

From 10 March 2017 to 30 June 2017, the Registry conducted the Health and Quality of Life Survey (HQoL). Eligibility for this survey included completing all four Registry survey waves, being at least 18 years of age, and speaking English. Two groups were invited by email or mail to participate in the survey. The first group (*n* = 2699) included persons who reported on Wave 1 that they sustained one or more of the following injuries on 9/11: cut, abrasion, or puncture wound; sprain or strain; burn; broken bone or dislocation; and concussion or head injury. Those who reported “other injury” or “eye injury” only were not included. The second group (*n* = 2598) consisted of a non-injured comparison group of randomly selected persons who did not report any type of injury, including “other injury” or “eye injury.” The overall participation rate was 76%, with a final sample of *n* = 2038 for the injury group and *n* = 1995 for the comparison group. For the purposes of the current study, we further restricted the sample to persons who were south of Chambers Street on 9/11/2001 (*n* = 2583 excluded) and who inconsistently reported that they were injured or not injured on 9/11 in both the Wave 1 survey and the HQoL (*n* = 1185 excluded). This resulted in *n* = 948 for the injured group and 1818 for the non-injured comparison of SF-12 outcomes [19].

### 2.2. SF-12 Outcomes 

The HQoL survey included a series of questions referred to as the Short Form Health Survey-12, Version 1 (SF-12) [20]. The SF-12 was derived from the SF-36 to provide an efficient method for assessing overall physical and mental health functioning through a mean physical health Component summary score (PCS-12) and a mean mental health component summary score (MCS-12). The summary scores are based on combinations of SF-36 questions that were identified as representing overall physical and mental status and that were highly correlated with the SF-36 overall physical and mental health summary scores. The SF-12 consists of physical health domains of *general health* (one question), *pain interferes with functioning* (one question), *role physical* (two questions, less able to perform routine tasks), and *overall physical functioning* (two questions). Mental health domains include *vitality* (one question), *social functioning* (one question), *mental health* (two questions), and *role emotional* (two questions) (less able to engage emotionally). Both of the SF-12 summary scores are highly correlated with SF-36 (*r* = 0.95) and have acceptable test-retest reliabilities (PCS-12 *r* = 0.89 and MCS-12, *r* = 0.76) [21].

### 2.3. Level of Injury Severity

An estimate of the level of injury severity was defined using follow-up questions on the HQoL questionnaire, which asks about the most serious injury sustained on 11 September 2001. Four categories were derived for this analysis, which are no injury, low severity, medium severity, and high severity. Low severity represents persons with superficial injuries not requiring medical intervention or care based on no reported need for medical care, use of support (e.g., crutch), or physical therapy. Medium severity applies to persons with injuries that required supportive and rehabilitative care, such as staying in bed for at least a day, requiring a crutch, or participating in physical therapy, but did not require using a wheelchair, going to a hospital, or surgery. High severity indicates that the injury required a hospital emergency department visit, surgery, or a wheelchair during recovery from the injury. The ‘no injury’ group included persons who reported no type of injury sustained on 9/11 on both the Registry Wave 1 and HQoL.

### 2.4. History of PTSD

We included the presence or absence of probable PTSD based on PCL scores of 44 or greater on any prior Registry survey waves (2003–2004, 2006–2007, 2010–2011, or 2015–2016). Probable PTSD was assessed in all Registry survey waves using a 17-item 9/11-specific PCL [22]. The 17 items correspond to the Diagnostic and Statistical Manual of Mental Disorders (DSM-IV) PTSD symptoms [23]. Each stressor-specific item, such as “feeling very upset when something reminded you of the events of 9/11,” was scored on a 5-point scale for experience of the symptom during the past 30 days (1 = not at all to 5 = extremely). The PCL score has been shown to have good temporal stability, internal consistency (>0.75), test-retest reliability (0.66), and high convergent validity [24], with overall diagnostic efficiency = 0.90, sensitivity = 0.94, and specificity = 0.86 [25]. 

### 2.5. Other Covariates 

Sociodemographic characteristics of the study sample such as gender, age at time of HQoL survey, race/ethnicity group, education level, eligibility group (rescue/recovery worker, resident, area worker, passerby), and marital status were included in the analytical models. Other covariates that have a potential association with physical and mental health functioning including smoking history (W4), social support, and self-efficacy were also controlled for in the analysis.

Because low levels of social support are associated with an increased burden of chronic PTSD [26], we included a measure of social support as a covariate. The presence of social support and/or self-efficacy described below also has a likely role in faster recovery from injury [27]. The instrument we used for measuring social support was the Medical Outcomes Study (MOS) Modified Social Support Survey [28,29]. Respondents were asked a series of questions about their support system, such as: “Is someone available to take you to the doctor if you need to go?”, “to have a good time with?”, “to hug you?”, “to prepare your meals if you are unable to do it yourself?”, and “to understand your problems?” Items were rated on “0” (none of the time) to “4” (all of the time) by the respondent. Out of a possible sum of 20 for the five questions, persons with scores <17 (median score for the entire sample) were assigned to a low social support category.

Self-efficacy, which is correlated with emotional health and optimism, was measured using five questions from the 10-item General Self-Efficacy Scale (GSE) [30]. The items were: “It is easy for me to stick to my aims and accomplish my goals”, “I am confident that I could deal efficiently with unexpected events”, “Thanks to my resourcefulness, I know how to handle unforeseen situations”, “I can remain calm when facing difficulties because I can rely on my coping abilities”, and “No matter what comes my way, I am usually able to handle it” [31]. The respondent rated each question using the following scale: 1 = not true at all, 2 = hardly true, 3 = moderately true, and 4 = exactly true. Scores ranged from 4 to 20 with a score equal to or greater than the median self-efficacy summary score of 17, which was used as the criterion for possessing self-efficacy. A variable representing diagnosed chronic conditions including asthma, heart disease, gastroesophageal reflux syndrome (GERS), and other non-neoplastic lung conditions was examined in relation to the outcomes in bivariate analyses. The variable was not controlled for in analytical models because these could have been on the causal pathway between injury and physical or mental health functioning [32].

### 2.6. Statistical Analysis

The primary outcomes of this analysis were physical and mental health summary component scores (PCS-12 and MCS-12). First, age-standardized mean PCS-12 and MCS-12 scores by gender and injury and PTSD status were compared with national data obtained from a study of a normative, non-institutionalized US sample [33]. The HQoL scores of those without injury or PTSD were age standardized to the age distribution (by 10-year age groups) provided by Hanmer et al., 2006 [33]. Internal comparisons were then done using persons with no injury and no PTSD as a comparison to persons with injury and no PTSD, persons with no injury and ever PTSD, and persons with injury and ever PTSD. Paired t-tests of PCS-12 and MCS-12 means at *p* < 0.05 level of significance were used to identify differences in these comparisons.

In addition, mean MCS-12 and PCS-12 scores and standard deviations were calculated for descriptive results stratified by injury severity levels and PTSD status as well as other covariates included in this study. Analysis of variance was used to determine if the means for each variable were significantly different in the bivariate analysis.

Multivariable analysis consisted of linear regression models of PCS-12 and MCS-12 scores with an injury severity level and PTSD ever and never included as primary predictors, controlling for age at the time of HQoL survey, gender, race/ethnicity, education, smoking status, eligibility group, marital status, social support, and self-efficacy. Negative or positive betas represented the amount of change in the PCS-12 or MCS-12 score relative to the referent for the factor of interest.

Additional analysis was done to assess the interaction between injury severity and PTSD in their association with PCS-12 and MCS-12 means. A cross-product interaction term for injury severity and PTSD status was included in linear regression models for PCS-12 and MCS-12. In addition, a composite variable (variable representing the cross-product term in the regression model) was also created that combined the four levels of injury severity and PTSD (ever/never) with no injury and never PTSD as the reference for the following groups: no injury and ever PTSD, low severity injury and ever PTSD, medium severity injury and ever PTSD, high severity injury and ever PTSD in order to represent the combined effects of level of injury severity and PTSD.

## 3. Results

The mean Physical Component Score (PCS-12) for the entire HQoL sample was 43.3. The Mental Component Score (MCS-12) was 46.6. However, when the means were restricted to the ‘no injury and never PTSD’ group, they provided an approximate comparison to population-level measures of PCS-12 and MCS-12 (such as those provided by Hanmer, 2006 [33]), which were based on a US nationally representative sample (Table 1).

Specifically, men in the HQoL ‘no injury and never PTSD’ group had an age standardized mean PCS-12 of 45.5 compared with 48.9 from Hanmer. For this same group, the age standardized mean MCS-12 for men from the HQoL study was 50.7 compared to 52.0 in the Hanmer sample [33]. Women had mean scores that were less than three points different between HQoL and Hanmer for both PCS-12 and MCS-12. The mean PCS-12 for ‘no injury and never PTSD’ for women was 44.1 compared with 47.0 for Hanmer and 49.4 for mean MCS-12 compared with 50.4 based on Hanmer estimates [33]. 

Within our sample, injury of any severity level without PTSD diminished physical health by five points for men (PCS-12 mean: 39.8 vs. 44.6, t = 6.6, *p* < 0.0001 for injury and never PTSD vs. for no injury and never PTSD, respectively) and two points for women (PCS-12 mean: 41.9 vs. 43.6, t = 2.2, *p* < 0.029 for injury and never PTSD vs. for no injury and never PTSD, respectively) (Table 1). However, injury without PTSD did not produce a significant difference for either men or women for mean MCS-12 compared with no injury and never PTSD. Among men and women without injuries, those with PTSD had lower mental and physical health scores than those without PTSD (for men: PCS-12: 40.8 vs. 44.4, t = 4.8, *p* < 0.0001, MCS-12: 39.7 vs. 52.4, t = 4.4, *p* < 0.0001, for women: PCS-12: 40.9 vs. 43.6, t = 3.7, *p* < 0.0001, MCS-12: 39.0 vs. 49.7, t = 11.9, *p* < 0.0001). The injury and ever PTSD group PCS-12 and MCS-12 mean scores for men and women had lower values than the other groups. 

Overall, 380 (13.7%) of study enrollees had high severity injuries on 9/11/2001 and 960 (36.8%) reported symptoms indicative of ever having PTSD (Table 2). Both PCS-12 and MCS-12 mean scores were lower for persons with more severe injuries (F = 105.6, *p* < 0.0001 for PCS-12: F = 53.9, *p* < 0.0001 for MCS-12) and a history of PTSD (F = 271.9, *p* < 0.0001 for PCS-12: F = 27.7, *p* < 0.0001 for MCS-12). Persons with high injury severity had the lowest PCS-12 mean score (37.0) compared with those with no injury (46.6) and those with ever PTSD had a lower MCS-12 mean score (38.4) compared with those with never PTSD (51.2).

With regard to demographic characteristics, the mean PCS-12 and MCS-12 scores for females were slightly lower than for males (Table 2), which indicates slightly worse physical and mental health status for females when compared with men. In addition, PCS-12 mean scores decreased with age (i.e., physical health status worsened with increasing age), while MCS-12 mean scores increased with age (i.e., mental health status was relatively better with increasing age). Hispanic individuals had the lowest mean PCS-12 score (39.5) compared with other race/ethnicity groups. Persons with less than a high school education also exhibited the lowest mean score for PCS-12 (38.5) when compared with those with a college degree or higher. The same patterns were seen with the mean MCS-12 scores with the exception of age for which individuals younger than 45 years of age had the lowest mean MCS-12 score (45.6). Among Registry eligibility groups, rescue/recovery workers had a lower PCS-12 mean score (38.7) relative to other registry groups, but there was no significant difference between the eligibility groups for MCS-12 mean scores (F = 1.98, *p* = 0.115). In addition, persons with low social support and low self-efficacy had lower PCS-12 and MCS-12 mean scores when compared with those with high social support and self-efficacy.

The estimated betas (β) represent the results of linear regression as either increases or decreases of PCS-12 and MCS-12 mean scores relative to a reference category for each characteristic after controlling for other factors (Figure 1 and Figure 2). PCS-12 mean scores were significantly lower for persons with medium or high severity injuries than for those with no injury (medium severity β = −4.4, 95% confidence interval (CI), −5.4 to −3.3, high severity β = −4.1 95% CI, −5.2 to −3.0). In contrast, there was no significant relationship between MCS-12 mean score and injury severity on 9/11 in multivariable models, which was similar to results from the bivariate analysis.

Persons who had a history of probable PTSD had significantly lower summary scores for both PCS-12 and MCS-12 when compared with those with no history of PTSD. However, the beta coefficients for PTSD predicting mental health functioning were three times greater than those for predicting physical health functioning (β = −9.7, 95% confidence interval (CI) −10.6 to −8.7 for MCS-12 and PTSD, β = −2.8, 95%, CI −3.6 to −2.0 for PCS and PTSD).

Linear regression estimates of PCS-12 or MCS-12 differences for other factors in Figure 1 and Figure 2 mirrored the results of bivariate comparisons for covariates represented in Table 2. For instance, persons with more than a high school education had significantly higher PCS-12 mean scores than those with less than a high school education, but there was no statistically significant relationship between education and mental health functioning. With regard to eligibility groups, rescue/recovery workers had significantly lower PCS-12 mean scores relative to residents, area workers, and passerby groups, but they had a significantly higher MCS-12 mean score (*p* < 0.001). Other notable findings for covariates included social support at Wave 4 that was predictive of mental functioning but not physical functioning, and self-efficacy significantly predicted both physical and mental health functioning. In addition, smoking history was not associated with PCS-12, but being a former smoker was a statistically significant predictor of lower MCS-12 mean scores (*p* < 0.001). Being diagnosed with a chronic disease prior to the assessment of physical and mental health functioning was not included in the model because it has been shown to be in the pathway for health, as measured by SF-12 [34]. In a sensitivity analysis, the inclusion of this measure in the model did not substantively alter the results.

The interaction between the injury severity level and probable PTSD on physical and mental health functioning was evaluated in a separate linear regression model in which the results are depicted in Figure 3 (PCS-12) and Figure 4 (MCS-12). Overall, although the interaction was not statistically significant, the combination of greater injury severity level and having a history of PTSD were associated with declines in the physical health status. However, injury severity had minimal influence on the association between PTSD history and mental health status. Specifically, people who had both a history of probable PTSD and medium or high severity injury had a substantially lower overall physical health status when compared with those with no injury and no PTSD (Figure 3). With regard to one’s mental health status, PTSD had a dominant impact on mental health functioning in comparison with any severity level of injury. Those with a history of PTSD (ever PTSD) had mean scores of up to 12 points lower when compared with those without PTSD regardless of the injury status or level (Figure 4).

## 4. Discussion

Based on prior research, we hypothesized that being injured on 11 September 2001 due to attacks on the World Trade Center would have long-term consequences on quality of life. In this study, we assessed both physical and mental health functioning using the well validated SF-12 health status instrument. The assessment demonstrated significant deficits for both the physical and mental health status among those who were injured on 9/11 and/or those with ensuing PTSD symptoms. For instance, there was a 7 to 10 point (women and men, respectively) deficit in SF-12 physical health summary score for those with an injury on 9/11 and no history of probable PTSD compared with population-based measures of the physical health status. Moreover, levels of injury severity, as defined by the degree of medical intervention following the 9/11 injury, had a dose-response association with a magnitude of the physical health function decrement but no association with mental health functioning. Thus, being injured on 9/11 is a major risk factor for long-term physical health effects among those directly exposed to the attacks, which represents a continued impact on a substantial population of affected persons.

Our finding that injury on 9/11 had an impact on physical functioning 15 years after sustaining an injury on 11 September 2001 indicates the extent to which an injury can diminish a person’s long-term capacity to function. Few other studies have assessed the functional health status after this length of time following an injury even though some have found significant deficits after shorter periods. For example, one study reported an 80% increased likelihood of reporting poor health among injured individuals when compared with non-injured individuals five years after the injury occurrence [9]. There are a number of plausible mechanisms for a disaster-related injury to have long-term effects on physical health. First, the physical damage of the injury could persist as chronic pain and interfere with normal activity. Second, an injured person may be vulnerable for developing chronic disease, such as cardiovascular disease, due to inactivity secondary to the injury [1,35]. Third, people who are injured are more likely to engage in behavioral risks (e.g., smoking, alcohol consumption, or lack of exercise), which can result in diminished physical health through a number of possible pathways [36]. However, we did not find an association between smoking and physical health, as measured by PCS-12.

A history of probable PTSD was also the primary factor associated with the decline of both physical and mental health functioning. Specifically, PTSD had a significant effect on physical functioning with a three-point decrement in a physical health composite score between those with ever PTSD versus those with no history of PTSD after controlling for other factors, including injury severity. However, there was a three-fold greater likelihood of a history of probable PTSD predicting adjusting for injury severity and other covariates. This is in accord with several studies that have assessed the impact of mental health stress on overall mental health using SF-12 domain mean scores. For instance, a study on veterans of the 1991 Gulf War reported a negative correlation between the level of military stressors and mean MCS, as measured by the SF-12 and Military Service Experience questionnaire [37]. Another study reported that the mental health checklist scores that assessed the presence of psychological disorders including PTSD, depression, or general anxiety disorder, which were significantly negatively correlated with MCS-12 [38].

Other studies have similarly reported that injury impacts physical health more than mental health [11]. Two explanations are that the presence of comorbid conditions is associated with reduced capacity for self-care among injured persons [12], or that chronic conditions fully mediate the impact of 9/11 exposure on physical health status and partially mediate mental health status, as measured by SF-12 [34]. However, models run in our sensitivity analyses indicated that the regression coefficients for injury or PTSD were not altered for PCS-12 means, regardless of whether the number of diagnosed conditions were included in the models, which indicates that a history of the measured chronic conditions was not necessary for the injury and physical health relationship. 

While injury severity was significantly related to a physical health decrement but not mental health decrement, a history of PTSD increased the likelihood of worse functioning for each level of injury severity for both physical and mental health. The obverse was evident when injury increased the PTSD-related decline in mental health. 

The findings from a prior qualitative study on the same population represented in this study suggested that being injured diminished the quality of life both physically and mentally regardless of PTSD status [15]. By using a much larger sample with standardized measures of physical and mental health functioning, we did not discern an independent relationship between injury severity and mental health functioning, but rather that PTSD symptoms were a dominant factor for overall mental health functioning, as measured by the SF-12.

Many of the associations between demographic characteristics and other factors (e.g., social support) with physical and mental health summary scores were similar to those that have been reported for other populations. For instance, physical health functioning declined with age, but not mental health status, as exemplified by a mean PCS-12 score of 49 for 30-year-olds to 44-year-olds and a mean score of 42 for those 65 years and older. Similarly, another study based on a large US national survey reported a PCS-12 mean score that was −0.85 less than the population norm for 30–39 year olds and −5.1 for those aged 65 to 69. However, there was an increase by age for MCS-12 mean scores from −0.12 less than the population norm for those aged 30 to 39 to +2.4 years for those aged 60 to 69 years [39]. Gopinath [14] also reported a −1.5 decline in PCS-12 mean for each additional 10 years of age, but no change for the mean MCS-12 based on persons with minor musculoskeletal injuries. With regard to gender, other studies have noted that men have higher mean scores on PCS-12 than women, but similar mean MCS-12 scores [33,39] consistent with results from this study. Although we did not specifically evaluate the association between PTSD history for men and women separately with mental health status, the finding that there were no differences between men and women is surprising because women generally have higher PTSD rates than men [40,41] 

With regard to other factors, Soberg [11] reported a three-fold higher level of physical and mental health summary scores within a year following injury for those with post-high school education compared with those with a high school-only education. However, in our study, we did not find a significant association between the educational level and MCS-12 or PCS-12. Other factors, such as social support or self-efficacy, were not typically included as factors in studies using PCS-12 or MCS-12. However, Kiely [42] reported that less social support among injured persons six months after injury was associated with a mental health deficit, as measured by the MCS-36, but not with physical health, as measured by the PCS-36. Our study had similar results.

The findings from this study indicate that both PCS-12 and MCS-12 are sensitive to long-term changes in health status and general health functioning many years after the disaster. The patterns of association between demographic and socio-behavioral factors concur with the magnitude of change in the PCS-12 and MCS-12 in various reports based on general population data and in studies specific to injured persons. The degree to which SF-12 is a strong indicator of health status is evident by the SF-12 having a dose-response relationship to a biologically based assessment of physical health referred to as a frailty index [43].

This study has several key strengths. First, we were able to assess physical and mental health status after a much longer period since the original event was compared with other studies. Second, we had a sufficient sample of persons injured on 11 September 2001 as well as a comparison group of non-injured persons to assess the combined influence of multiple characteristics longitudinally on physical and mental health status.

However, the study is also subject to some limitations. First, there is likely self-selection bias on a number of dimensions. For example, we only included persons who had participated in all of the registry follow-up surveys. A large proportion of persons in the cohort were also self-selected for original enrollment in the registry. An assessment of the impact of follow-up survey participation in the registry found that an increased propensity to participate in surveys was related to an absence of chronic conditions [17]. However, the overall physical and mental health of those who had no injury and never had PTSD in this study were not substantively different from the general US population. Second, we relied on self-report, which is especially problematic for key predictors in this study including having been injured on 9/11 and probable PTSD. Nonetheless, we defined being injured and injury severity in a way that minimized the bias of self-report by eliminating persons who provided inconsistent responses concerning their injury on 9/11 between what was reported at the registry enrollment survey and what was reported in the HQoL survey. Lastly, we do not have information on psychiatric conditions or vulnerabilities prior to the 9/11 trauma, which would have permitted controlling for this factor in the association of PTSD history with physical and mental health status.

## 5. Conclusions

In this study, we documented convincing evidence that many persons injured on 11 September 2001 were experiencing diminished physical health 15 years after the event. To put it into context, the mean PCS-12 score for persons injured without PTSD is comparable to reported mean PCS-12 scores for those with cerebral aneurysms (39.5) or congestive heart failure (42.8) [44,45]. Given that many persons who sustained an injury on 11 September 2001 also subsequently suffered from symptoms of PTSD, we found that, after adjusting for other factors, the combined effect of severe injury with PTSD yielded significant deficits in both physical and mental health functioning when compared with those who were not injured and did not experience PTSD. Even though, over the course of years since the injury, the physical wounds of the injury could have healed in some cases, our results suggest that traumatic injuries appear to have a lasting effect on perceived physical functioning. The discontinuity between adequate physical functioning and significantly lower quality of life is a phenomenon that has been observed among injured persons and could occur in this population [39] even though we do not have evidence that directly supports this relationship. Nonetheless, the results from this study can be generalized to the long-term health burden at both the individual and societal level from nonfatal injuries sustained in natural or man-made disasters.

## Figures and Tables

**Figure 1 ijerph-16-01054-f001:**
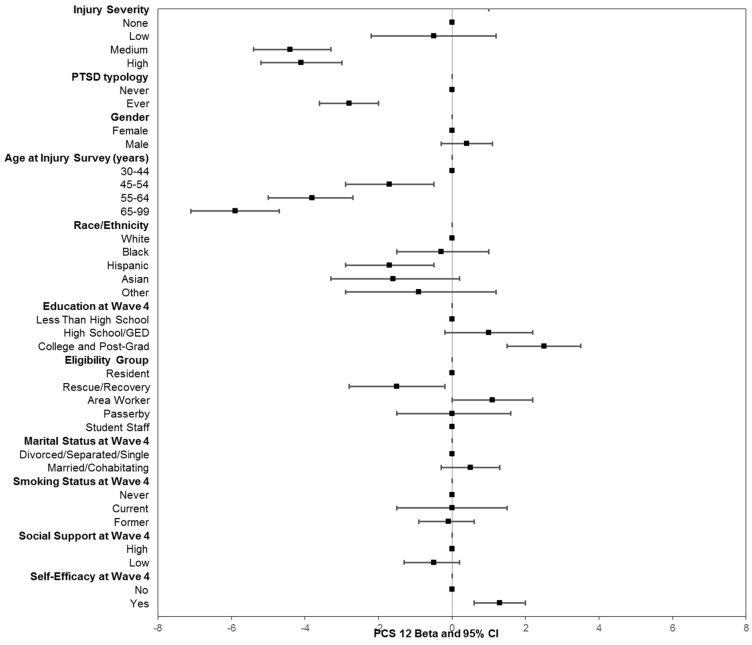
Linear regression of PCS-12 (physical health functioning) as a function of injury severity, PTSD history, demographic, and other factors. The betas are the predicted changes in mean scores relative to the reference category. Controlled for all variables included in figure. PTSD check list score ≥44 on any prior WTCHR survey.

**Figure 2 ijerph-16-01054-f002:**
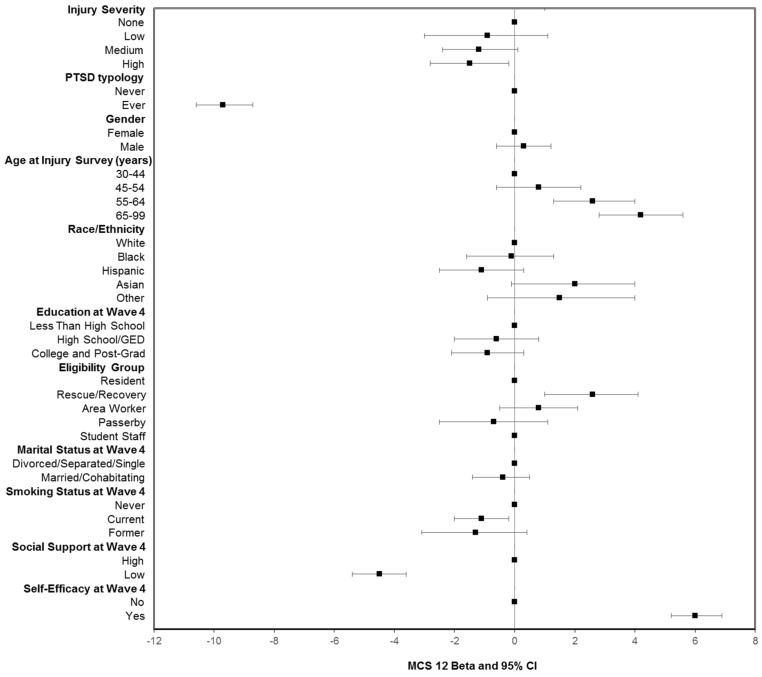
Linear regression of MCS-12 (mental health functioning) as a function of injury severity, PTSD history, demographic, and other factors. The betas are the predicted changes in mean scores relative to the reference category. Controlled for all variables included in the figure. PTSD checklist score ≥44 on any prior WTCHR survey.

**Figure 3 ijerph-16-01054-f003:**
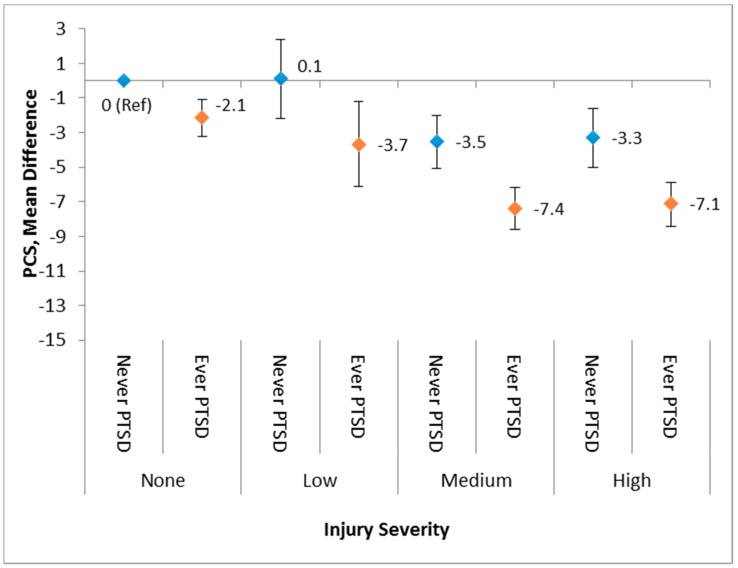
Adjusted regression beta coefficients for PCS-12 predicted by a combination of PTSD and the injury severity level.

**Figure 4 ijerph-16-01054-f004:**
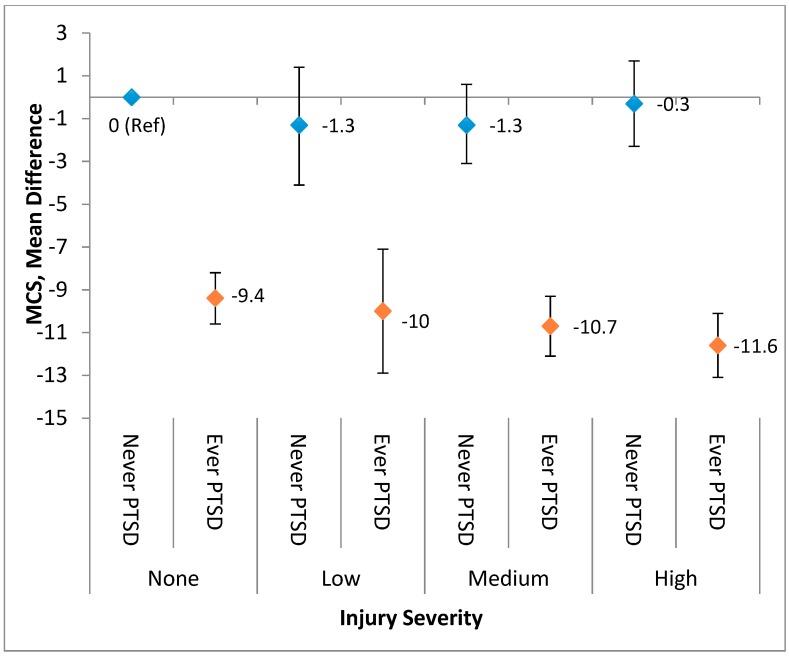
A combination of PTSD history and the injury severity level predicted adjusted regression beta coefficients for MCS-12.

**Table 1 ijerph-16-01054-t001:** Mean PCS-12 and MCS-12 for the US population and mean PCS-12 and MCS-12 for selected HQoL groups by gender.

Source	Male			Female		
	*N*	PCS-12	MCS-12	*N*	PCS-12	MCS-12
Non-institutionalized US adults *	7463	48.9	52.0	8819	47.0	50.4
HQoL: No injury and never PTSD **	768	45.5	50.7	589	44.1	49.4
HQoL: No injury and never PTSD ***	768	44.4	52.4	589	43.6	49.7
HQoL: Injury and never PTSD	200	39.8	52.5	95	41.4	48.8
HQoL: No injury and ever PTSD	156	40.8	39.7	198	40.9	39.0
HQoL: Injury and ever PTSD	343	34.8	38.5	263	35.4	36.9

* [33] ** Age standardized based on age distribution from Hanmer, 2006. *** Underlined are Non-age standardized for a comparison.

**Table 2 ijerph-16-01054-t002:** Mean SF-12 Physical Health Component Scores (PCS-12) and mean Mental Health Component Scores (MCS-12) by injury severity, PTSD history, and other covariates.

Characteristic	Total		Physical Functioning Total		Mental Functioning Total	
	*N*	%	Mean	SD	Mean	SD
Total Sample	2766	100	41.3	9.2	46.9	12.1
Injury Severity *						
None	1818	65.7	43.6	8.2	48.8	11.2
Low	120	4.3	40.9	10.2	45.6	13.1
Medium	448	16.2	37.0	9.1	42.5	12.7
High	380	13.7	37.0	8.7	42.1	12.8
PTSD **						
Ever	960	36.8	37.8	8.8	38.6	11.5
Never	1652	63.2	43.7	8.4	51.2	9.9
Gender						
Male	1547	55.9	41.6	9.2	47.9	12.0
Female	1219	44.1	41.7	8.7	45.5	12.1
Age at Injury Survey (Years)						
30–44	313	11.3	45.7	6.8	45.6	11.9
45–54	708	25.6	43.0	9.0	45.6	11.9
55–64	935	33.8	40.8	9.2	46.5	12.2
65–99	810	29.3	39.6	8.7	49.1	11.8
Race/Ethnicity						
White	2056	74.3	42.1	9.0	47.4	12.1
Black	261	9.4	40.9	8.9	45.6	12.5
Hispanic	264	9.5	39.5	8.9	43.6	11.6
Asian	109	3.9	41.0	8.7	48.1	12.4
Other	76	2.8	40.3	9.1	46.5	11.4
Education at Wave 1						
High School or Less	407	14.7	38.5	8.6	45.1	13.1
Some College	566	20.5	39.4	9.1	45.5	12.6
College and Post-Grad	1787	64.7	42.9	8.7	47.7	11.7
Eligibility Group						
Rescue/Recovery	626	22.6	38.7	9.6	47.4	12.3
Resident	317	11.5	42.6	8.1	46.5	12.2
Area Worker	1591	57.5	42.6	8.7	47.1	11.9
Passerby	232	8.4	41.6	9.1	44.6	12.5
Marital Status at Wave 4						
Married/Cohabitating	1873	68.4	42.0	9.0	47.8	11.8
Divorced/Separated/Widowed/Never Married	867	31.6	40.9	8.9	44.7	12.6
Social Support at Wave 4						
Low	1191	44.1	40.4	9.2	42.2	12.5
High	1512	55.9	42.6	8.7	50.4	10.5
Self-Efficacy at Wave 4						
Yes	1189	43.5	43.3	8.7	52.4	9.8
No	1544	56.5	40.4	9.0	42.5	12.0
Smoking—Wave 4						
Current	159	5.8	40.1	9.0	43.4	11.5
Former	853	31.3	41.3	9.0	46.8	12.4
Never	1713	62.9	42.0	9.0	47.3	12.0
Any Chronic Disease—Wave 4 ***						
Yes	211	8.6	37.8	9.0	44.2	13.3
No	2241	91.4	42.0	8.9	47.1	12.0

* Severity was missing on 55 individuals due to missing data on medical intervention after injury, ** PTSD checklist score ≥44 on any prior WTCHR survey, *** Diagnosed chronic conditions included asthma, heart disease, GERS, and non-neoplastic lung conditions.

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
