# Peer review of "An Assessment of Long-Term Physical and Emotional Quality of Life of Persons Injured on 9/11/2001"

_ijerph, 2019, doi:10.3390/ijerph16061054_

Round 1

Reviewer 1 Report

This is an interesting paper, focused on the relationship between PTSD and physical injuring with respect to long-term quality of life after 9/11. The field is worth investigation and the methods employed by the authors seem to be adequate to the purpose. The article is overall well written, information about methods and results is clear and in the discussion section authors adequately discuss their results in light of previous literature. However, there are some points that could be improved in order to better contestualize the study in the current literature about PTSD.

In particular, given the wide literature that addressed the issue of gender (generally reporting higher rates in women) and of age in development of PTSD (see e.g. PMID:23098636; PMID:24709023; PMID:30658014), authors should provide more insight on this matter in discussing their results, considering also the crucial role of PTSD in long-term quality of life reported in the study. Moreover, I think that another issue that should have been considered and adequately discussed is the presence of previous psychiatric conditions/personal vulnerabilities when subjects were exposed to the traumatic experience, with or without physical injuries. This is of crucial importance given the current debate about vulenrability/resiience factors for developing PTSD, which features even the possible development of specific subtypes of PTSD also in response to milder stressful events in high-risk subjects (see e.g. PMID:30781888; PMID:30366638). If this data is not available in detail, I think that implications should be discussed in the limitation section of the study.  

Author Response

Comments and Suggestions for Authors from Reviewer 1

This is an interesting paper, focused on the relationship between PTSD and physical injuring with respect to long-term quality of life after 9/11. The field is worth investigation and the methods employed by the authors seem to be adequate to the purpose. The article is overall well written, information about methods and results is clear and in the discussion section authors adequately discuss their results in light of previous literature.

Response:  The authors thank the reviewer for their positive comments.

However, there are some points that could be improved in order to better contestualize the study in the current literature about PTSD.In particular, given the wide literature that addressed the issue of gender (generally reporting higher rates in women) and of age in development of PTSD (see e.g. PMID:23098636; PMID:24709023; PMID:30658014), authors should provide more insight on this matter in discussing their results, considering also the crucial role of PTSD in long-term quality of life reported in the study.

Response: Per reviewers suggestion we added two references Bowler (2012) “Longitudinal mental health impact among Police responders to the 9/11 Terrorist Attack (Am J Ind Med) and Gogos (2019) “Sex differences in schizophrenia, bipolar disorder, and post-traumatic stress disorder:  Are gonadal hormones the link” (Br J Pharmacol).  We added text on how the difference by gender for PTSD was not reflected in the results (line 386-389).

Moreover, I think that another issue that should have been considered and adequately discussed is the presence of previous psychiatric conditions/personal vulnerabilities when subjects were exposed to the traumatic experience, with or without physical injuries. This is of crucial importance given the current debate about vulenrability/resiience factors for developing PTSD, which features even the possible development of specific subtypes of PTSD also in response to milder stressful events in high-risk subjects (see e.g. PMID:30781888; PMID:30366638). If this data is not available in detail, I think that implications should be discussed in the limitation section of the study.  

Response: We appreciate the reviewer’s insight into the role of prior psychiatric conditions on the impact of PTSD.  We agree that personal vulnerabilities and the presence of psychiatric conditions are important considerations in understanding the course of PTSD following a trauma, but we are not clear whether including history of psychiatric conditions would be pertinent to the results of this study.  The goals of the study were to assess the level of physical and mental health functioning 15 years after the initial disaster exposure and trauma, given whether the persons sustained a physical injury as a result or had subsequent PTSD. We might posit that prior history of psychiatric problems could independently from PTSD influence future physical and mental functioning but this is different type of analysis then our study objectives.  There are, in any case, no available data for evaluating this pathway.  Nonetheless, we added the absence of this information on prior history to limitations section of discussion (line 421-424).

Reviewer 2 Report

This is an important study using longitudinal data from the WTC post-disaster registry that was initiated in 2002 and has data from 71,426 persons exposed to the attacks on 9/11/2001. The mission of this study is to better understand the impact of this atrocity on the estimated populations-at-risk of 409,492 
persons. This report adds to the understanding of long-term physical and mental health consequences of 9/11. Since physical injury and mental health conditions like Post-Traumatic Stress Disorder symptoms (PTSD) occur together in responders or area workers and residents this report describes  the association of early physical injury and symptoms of PTSD on self reported physical and mental health 15 years after exposure. The data convincing demonstrate that early physical injury worsen late PCS-12 and that ever PTSD contribute to lower PCS-12 in those with medium or high intensity injury. As expected PTSD had a stronger effect on self reported mental health (MCS) than on physical health.

My major critique has to do with statements like “Although physical wounds of the injury likely 
healed.” and  “Over the course of years since the injury, we might 
 expect that the physical wounds of the injury likely healed. ” These suppositions are not supported by other sections of the discussion such as “the physical damage of the injury could persist as chronic pain and interfere with normal 
 activity ”.  

The discussion could be more cohesive . For example the description of data from this report  “The positive relationship 
 between age and mental health status suggests that mental health is less affected by health decline 
 that results from physical injury. 
” is in a separate paragraph from the  description of prior literature “physical health functioning declined with age, but not 
 mental health status ” 

Finally the large amount of data in Table 3 would be better presented visually such as by a forest plot on Mean and CI of PCS and MCS by each characteristic.

Author Response

Comments and Suggestions for Authors from Reviewer 2

This is an important study using longitudinal data from the WTC post-disaster registry that was initiated in 2002 and has data from 71,426 persons exposed to the attacks on 9/11/2001. The mission of this study is to better understand the impact of this atrocity on the estimated populations-at-risk of 409,492 
persons. This report adds to the understanding of long-term physical and mental health consequences of 9/11. Since physical injury and mental health conditions like Post-Traumatic Stress Disorder symptoms (PTSD) occur together in responders or area workers and residents this report describes  the association of early physical injury and symptoms of PTSD on self reported physical and mental health 15 years after exposure. The data convincing demonstrate that early physical injury worsen late PCS-12 and that ever PTSD contribute to lower PCS-12 in those with medium or high intensity injury. As expected PTSD had a stronger effect on self reported mental health (MCS) than on physical health.

Response: The authors appreciate the positive comments by the reviewer.

My major critique has to do with statements like “Although physical wounds of the injury likely 
healed.” (abstract) and  “Over the course of years since the injury, we might 
 expect that the physical wounds of the injury likely healed. ” (conclusion)These suppositions are not supported by other sections of the discussion such as “the physical damage of the injury could persist as chronic pain and interfere with normal 
 activity ”.  

Response:  The authors thank the reviewer for identifying some confusing statements in the discussion.  We have revised the last sentence in the abstract to say “Although it is unknown whether physical wounds of injury healed, our results suggest that traumatic injuries appear to have a lasting negative effect on perceived physical functioning” (lines 25-27) Also, modified the sentence in the Conclusion to say “Although over the course of years since the injury, the physical wounds of the injury could have healed in some cases, our results suggest that traumatic injuries nonetheless appear to have a lasting effect on perceived physical functioning.” (lines 433-435) With these revisions there should be no contradiction between statements in the discussion.

The discussion could be more cohesive . For example the description of data from this report  “The positive relationship 
 between age and mental health status suggests that mental health is less affected by health decline 
 that results from physical injury. 
” is in a separate paragraph from the  description of prior literature “physical health functioning declined with age, but not 
 mental health status ” 

Response:   Thanks pointing out the misplacement of one of these statements in the discussion.  The authors determined that the sentence on lines 362-364 could be deleted without losing information in the discussion.

Finally the large amount of data in Table 3 would be better presented visually such as by a forest plot on Mean and CI of PCS and MCS by each characteristic. Response:  We replaced Table 3 with forest plot figures one for physical health and the other for mental health and modified the text accordingly.   
